# (L)-Monomethyl Tyrosine (Mmt): New Synthetic Strategy via Bulky ‘Forced-Traceless’ Regioselective Pd-Catalyzed C(sp^2^)–H Activation

**DOI:** 10.3390/ph16111592

**Published:** 2023-11-10

**Authors:** Davide Illuminati, Claudio Trapella, Vinicio Zanirato, Remo Guerrini, Valentina Albanese, Chiara Sturaro, Simona Stragapede, Davide Malfacini, Greta Compagnin, Martina Catani, Anna Fantinati

**Affiliations:** 1Department of Life Sciences, University of Modena and Reggio Emilia, Via G. Campi 213/d, 41125 Modena, Italy; davide.illuminati@unife.it; 2Department of Chemical and Pharmaceutical Sciences, University of Ferrara, Via Fossato di Mortara 17, 44121 Ferrara, Italy; trap@unife.it (C.T.); znv@unife.it (V.Z.); grm@unife.it (R.G.); greta.compagnin@unife.it (G.C.); martina.catani@unife.it (M.C.); 3Department of Environmental and Prevention Sciences, University of Ferrara, 44121 Ferrara, Italy; valentina.albanese@unife.it; 4U.O. Neurological Clinic, University Hospital of Ferrara, Via Aldo Moro, 8, 44124 Ferrara, Italy; chiara.sturaro@unife.it (C.S.);; 5Department of Pharmaceutical and Pharmacological Sciences, University of Padova, Via 8 Febbraio, 2, 35131 Padova, Italy; davide.malfacini@unipd.it

**Keywords:** monomethyltyrosine, dimethyltyrosine, catalysis, peptides

## Abstract

The enormous influence in terms of bioactivity, affinity, and selectivity represented by the replacement of (L)-2,6-dimethyl tyrosine (Dmt) instead of Phenylalanine (Phe) into Nociceptin/orphanin (N/OFQ) neuropeptide analogues has been well documented in the literature. More recently, the non-natural amino acid (L)-2-methyl tyrosine (Mmt), with steric hindrance included between Tyr and Dmt, has been studied because of the modulation of steric effects in opioid peptide chains. Here, we report a new synthetic strategy to obtain Mmt based on the well-known Pd-catalyzed *ortho*-C(sp^2^)–H activation approach, because there is a paucity of other synthetic routes in the literature to achieve it. The aim of this work was to force only the mono-*ortho*-methylation process over the double *ortho*-methylation one. In this regard, we are pleased to report that the introduction of the dibenzylamine moiety on a Tyr aromatic nucleus is a convenient and traceless solution to achieve such a goal. Interestingly, our method provided the aimed Mmt either as *N*-Boc or *N*-Fmoc derivatives ready to be inserted into peptide chains through solid-phase peptide synthesis (SPPS). Importantly, the introduction of Mmt in place of Phe^1^ in the sequence of N/OFQ(1-13)-NH_2_ was very well tolerated in terms of pharmacological profile and bioactivity.

## 1. Introduction

(L)-2,6-dimethyl tyrosine (Dmt) and the corresponding monomethyl tyrosine (Mmt), have been widely used as tyrosine analogues introducing steric encumbrance in bioactive peptides. In particular, Dmt amino acid residue significantly increased bioactivity, affinity, and selectivity of synthetic opioid peptides containing it in place of native Tyr or Phe amino acids [1,2,3,4,5,6,7,8,9,10,11,12,13,14,15,16,17,18].

At the current state, while a series of synthetic pathways have been developed for the synthesis of Dmt [1,19,20,21,22,23,24,25], there are only few methods for the obtainment of Mmt [26,27,28].

Thus, in 1991, McDonald et al. reported the synthesis of racemic Mmt by the alkylation of the anion of diethyl acetamido malonate with 4-bromomethyl-3-methyl anisole [29]. Later on, Lazarus and Okada **[30]** achieved 2-monomethyl tyrosine through palladium-catalyzed coupling of a suitably trisubstituted iodophenol with 2-acetamido acrylate. In the subsequent step, the resulting sterically hindered dehydroamino acid was subjected to enantioselective hydrogenation in order to establish the desired C-*alpha* configuration [1]. An elegant synthetic route to Mmt was proposed by Majer et al. [31] who conceived to obtain a regioselective monomethylation of Tyr via hydrogenolysis of tetrahydroisoquinoline-3-carboxylic acid prepared from Tyr, through a Pictet–Spengler cyclization. Unfortunately, the drastic conditions required in the reductive step led to extensive racemization leading to the obtainment of Mmt in racemic form. More recently, Mosberg et al. described a 3-steps synthesis to achieve non-natural Tyr derivatives in low to discrete yields, through a Negishi cross-coupling between an NH-Boc-Iodoalanine-OMe derivative and 3-methyl-4-iodophenol, using Pd_2_(dba)_3_ and SPhos [24] (see Figure 1).

In this work, we describe the successful preparation of the targeted non-natural (L)-amino acid Mmt through a synthetic pathway inspired by our recently reported strategy for the synthesis of a small library of 4-substituted Dmt-like amino acids (Figure 2) [32]. In our previous work, indeed, we established that the dibenzylamine functional group (Bn_2_N) was well tolerated in the pivotal Pd-catalyzed *ortho*-C(sp^2^)–H activation-methylation sequence carried out in accordance with Zhang and Ma [25]. In addition, Bn_2_N showed to be a convenient precursor for a list of functional groups that were subsequently introduced by using a classical *ipso*-substitution reaction of the corresponding diazonium salt. As anticipated, by using picolinamide (PA) as a directing group and methyl iodide as the alkylating reagent, it was not possible to kinetically differentiate methylation at C-2 from the one at C-6, so that a double *ortho*-dimethylation of the aryl propionic acid derivative was the only recorded event. We were confident that by performing the Pd-catalyzed *ortho*-activation-methylation sequence on a sterically constrained aryl propionic acid derivative, the methylation reaction would occur exclusively at the most accessible aromatic *ortho*-position. The previously acquired knowledge suggested that we turn to the Bn_2_N group as a traceless regioselective inductor because it is undoubtedly sterically demanding, chemically tolerant, and easily removable.

Although not yet tested as extensively as Dmt, the Mmt analogue appears to be an interesting amino acid residue to be inserted in peptide chains because of some promising biological evaluations already reported for MOP-selective agonists. In particular, [Mmt^1^]DALDA (H-Tyr-D-Arg-Phe-Lys-NH_2_) [4] and [Mmt^1^]EM-2 (endomorphin-2) [30] have shown increased MOP (mu opioid receptor) receptor-binding affinity and MOP agonist activity compared to the corresponding parent peptides. Furthermore, similar receptor selectivity has been confirmed for Mmt^1^ and Dmt^1^ peptide derivatives, with the latter being more potent. Herein, we also report some preliminary results on [Mmt^1^]N/OFQ(1-13)-NH_2_, the shortest active fragment of N/OFQ (i.e., N/OFQ(1-13)-NH_2_) in which the synthesized Mmt has been inserted at the *N*-terminal amino acid residue. Effectively, in vitro pharmacological evaluation displayed highly interesting results confirming that Mmt represents an intriguing target in terms of innovative neuropeptides.

## 2. Results and Discussion

### 2.1. Chemistry

As reported by Ma and Zhang [25], the picolinamide functional group acts as a suitable *ortho*-directing group in the key step of Pd-catalyzed C(sp^2^)–H activation-methylation. Consequently, our synthesis (Figure 3) started with the methyl esterification of the commercially available 3-nitro-L-tyrosine **1** in order to allow its binding to picolinic acid via the HATU–DIPEA condensation system. The resulting *N*-protected tyrosine derivative **2** was transformed to the corresponding TBDMS ether **3** before the conversion of the nitro group to the aimed Bn_2_N group. To this end, compound **3** was submitted to catalytic hydrogenation and the resulting primary arylamine **4** was *N*-alkylated with benzyl bromide to give compound **5**. The latter was reacted with MeI in the presence of Pd(OAc)_2_ according to the already tested conditions [25,32]. Gratifyingly, the expected mono-*ortho*-methylated compound **6** was isolated in a 90% yield, confirming that the Bn_2_N moiety really prevents the methylation reaction from taking place at the adjacent *ortho*-position. The subsequent hydrogenolytic step removed both benzyl groups from the arylamine **6** affording compound **7** from which Boc–Mmt **10** or Fmoc–Mmt **11** could be promptly obtained in moderate to good yields through the sequence of protodeamination, hydrolytic protective group removal, and classical *N*-protection. In detail, the reaction of compound **7** with the isopentyl nitrite–fluoroboric acid system gave the corresponding aryl diazonium salt **8,** which was protodeaminated to **9** by treatment with FeSO_4_ in DMF. The following harsh acidic hydrolytic conditions yielded Mmt hydrochloride salt, once treated with Boc_2_O afforded **10** or, alternatively treated with FmocCl afforded compound **11**. (L)-Mmt **10** was used for the synthesis of [Mmt^1^]N/OFQ(1-13)-NH_2_, through SPPS, following a previously described protocol.

The reported synthetic strategy to Mmt widened the scope of the original stereoconservative Pd-catalyzed *ortho*-C(sp^2^)–H activation-methylation of Tyr, already described to prepare Dmt [25].

It has been verified that the Bn_2_N moiety judiciously introduced on the Tyr aromatic nucleus is a convenient traceless functional group to obtain a single Pd-catalyzed *ortho*-methylation, over the uncontrolled *ortho*-dimethylation previously reported, see Figure 4. The overall process allowed the preparation of chiral nonracemic Mmt suitably *N*-protected for use in SPPS.

### 2.2. Pharmacology

Here, we report the concentration–response curves of N/OFQ, N/OFQ(1-13)-NH_2_, [Mmt^1^]N/OFQ(1-13)-NH_2_, and [Dmt^1^]N/OFQ(1-13)-NH_2_ in the calcium mobilization assay by taking advantage of CHO_hNOP_ cells expressing the chimeric G protein G_αqi5_, which allows the coupling of the otherwise Gi/o-coupling NOP receptor to Gq–calcium. In this assay, all compounds evoked a concentration-dependent stimulation of calcium release. N/OFQ (1-13)-NH_2_ shows similar potency to and maximal effects (pEC_50_ = 9.82; E_max_ = 358) of the natural peptide N/OFQ (pEC_50_ = 9.56; E_max_ = 335). The substitution of Phe^1^ with Dmt in [Dmt^1^]N/OFQ(1-13)-NH_2_ produced a pEC_50_ of 8.35, showing a 30-fold reduction in potency compared to the parental peptide. Importantly, [Mmt^1^]N/OFQ(1-13)-NH_2_ mimicked the effects of the standard, with a pEC_50_ of 9.47. Of note, the statistical comparison of N/OFQ (1-13)-NH_2_ and the other ligands’ maximal effects indicated no significant alteration.

The concentration–response curves obtained with these ligands in CHO_hNOP_ cells are displayed in Figure 5, and their values of potency and maximal effects are summarized in Table 1.

The present study was carried out with the aim of evaluating the in vitro pharmacological activity of the new peptide [Mmt^1^]N/OFQ (1-13)-NH_2_ for its propensity to activate the human NOP receptor. The effects of [Mmt^1^]N/OFQ (1-13)-NH_2_, the previously reported [Dmt^1^]N/OFQ(1-13)-NH_2_ [8], the parental peptide N/OFQ(1-13)-NH_2_, and the endogenous neuropeptide N/OFQ were characterized via the calcium mobilization assay in the presence of chimeric G proteins. Intriguingly, in this assay, the [Mmt^1^]N/OFQ (1-13)-NH_2_ potency was higher than that of the Dmt derivative and similar to that of N/OFQ(1-13)-NH_2_ and N/OFQ. The design of NOP opioid peptides was based on seminal structure–activity relationship (SAR) studies on N/OFQ, which demonstrated that (i) the amidation of the C-terminal protects the peptide from enzymatic degradation; (ii) N/OFQ(1–13)-NH_2_ is the minimal bioactive sequence, which has been subsequently used as a chemical template for further SAR studies. In particular, a large series of N/OFQ(1–13)-NH_2_ derivatives were generated to investigate the importance of Phe^1^ for ligand efficacy and selectivity: Phe^1^ has an important role in the interaction with the receptor, and its substitution leads to changes in affinity [8]; indeed, the substitution with Phe^1^ in N/OFQ(1–13)-NH_2_ sequences causes a reduction in selectivity for NOP over mu opioid receptors; the substitution of Tyr^1^ with Dmt^1^ displayed a slight decrease in NOP potency associated with an increase in potency at the mu opioid receptor [8]. Even though Mmt was not tested as extensively as Dmt in position 1, previous reports demonstrate that Mmt is an interesting nonproteinogenic amino acid. In particular, the work of Tingyou Li and colleagues [30] evaluated the opioid receptor-binding affinities of EM-2 (endomorphin-2) analogs, and [Mmt^1^]EM-2 showed increased biological activity compared to the corresponding parent peptides when compared to [Dmt^1^]EM-2. In the present experiments, the peptides were evaluated for their capability to stimulate the release of intracellular calcium. To do that, CHO cells stably coexpressing the NOP opioid receptors and the Gα_qi5_ protein were used. The chimeric G protein is able to activate the PLC–IP–Ca^2+^ pathway, thus increasing intracellular calcium concentrations upon activation of an otherwise G_i/o_-coupling GPCR [33]. The calcium mobilization assay used for screening the NOP receptor ligands has been validated in previous studies. In particular, the pharmacological profile of the human NOP receptor coupled with calcium signaling has been assessed with large panels of well-known [34] and novel [8,35,36,37,38,39,40,41,42,43] ligands. In brief, the pharmacological profile obtained at the NOP receptor with this assay is in line with that of other more classical paradigms. Here, the N/OFQ(1-13)-NH_2_, N/OFQ, and [Dmt^1^]N/OFQ(1-13)-NH_2_ effects were in line with those reported in the literature, with N/OFQ(1-13)-NH_2_ and N/OFQ sharing similar values of potency and [Dmt^1^]N/OFQ(1-13)-NH_2_ being approximately 30-fold less potent [8,40]. Very importantly, the introduction of Mmt in place of Phe^1^ in the N/OFQ(1-13)-NH_2_ sequence was very well tolerated (equipotent to N/OFQ) and, if compared to the Dmt modification, the occurring [Mmt^1^]N/OFQ (1-13)-NH_2_ was 10-fold more potent. 

The results obtained for [Mmt^1^]N/OFQ (1-13)-NH_2_ might be enlightening for a spatial disposition within the NOP receptor-binding pocket close to that of the endogenous peptide. Further molecular dynamics studies could be of interest in validating this hypothesis. This has now been made possible by the recently reported high-resolution active structure of the human NOP receptor in a complex with N/OFQ and Gi [44].

## 3. Material and Methods

### 3.1. Chemistry

All commercial materials were purchased from Fluorochem and Sigma-Aldrich and used as received unless otherwise noted. Pd(OAc)_2_ (>98%, Fluorochem) was used in the Pd-catalyzed reactions. Instruments: Analytical RP-HPLC analyses were performed on a Xbridge^®^ C18 column (4.6 × 150 mm, 5 μm particle size) with a flow rate of 0.5 mL/min using a linear gradient of acetonitrile (0.1% TFA) in water (0.1% TFA) from 0% to 100% in 25 min. Retention times (Tr) from analytical RP-HPLC were reported in minutes. When necessary, compounds were purified on a reverse-phase Waters Prep 600 HPLC system equipped with a Jupiter column C18 (250 × 30 mm, 300 Å, 15 µm spherical particle size). Gradients used consisted of A (H_2_O + 0.1% TFA) and B (40% H_2_O in CH_3_CN + 0.1% TFA) at a flow rate of 20 mL/min. UV detection wavelength for semipreparative HPLC was 220 nm. Chiral HPLC analysis was performed on Whelk01 R,R 150 × 4.6 mm column packed with 3.5 µm particles size. Isocratic elution at 20% CH_3_CN and 80% H2O (0.1% TFA). Flowrate = 1 mL/min, injection volume = 20 µL, and temperature = 25 °C. All final products showed a degree of purity >95% at 220 and 254 nm. The mass spectra were recorded with a MICROMASS ZMD 2000. TLC was performed on precoated plates of silica gel F254 (Merck, Darmstadt, Germany). ^1^H NMR and ^13^C, DEPT NMR analyses were obtained at ambient temperature using a Varian 400 MHz spectrometer and were referenced to residual ^1^H signals of the deuterated solvents (δ ^1^H 7.26 for CDCl_3_, δ 1H 2.50 for DMSO-d6, and δ ^1^H 3.31, 4.87 for CD_3_OD); the following abbreviations were used to describe the shape of the peaks: s: singlet; d: doublet; dd: double doublet; t: triplet; and m: multiplet. Optical rotations were obtained on a Jasco P-2000 Polarimeter instrument with a path length of 1 dm (589 nm) and reported as follows: ${\left[\alpha \right]}_{T}^{D}$ (c = g/100 mL, solvent). The infrared analyses were performed with a spectroscopy FT-IR spectrum 100 (Perkin Elmer Inc., Waltham, MA, USA). Hydrogenation reaction in AcOEt was performed under continuous-flow conditions in an H-Cube Pro™ setup (Thalesnano, Hungary) equipped with a module for automatic control of operational parameters (reaction temperature in °C, pressure in bar, and flow rates of liquid feed in mL/min and hydrogen). Reactions with microwave assistance were performed using a Biotage Initiator TM 2.0 apparatus (Biotage, Vimpelgatan 5, 753 18 Uppsala, Sweden).

### 3.2. Pharmacology

Drugs and reagents: The reagents used were purchased from Sigma-Aldrich (St. Louis, MO, USA) and Thermo Fisher Scientific (Waltham, MA, USA). Peptides N/OFQ, N/OFQ(1-13)-NH_2_, [Mmt^1^] N/OFQ(1-13)-NH_2_, and [Dmt^1^]N/OFQ(1-13)-NH_2_ were synthesized at the Department of Chemical, Pharmaceutical and Agricultural Sciences of the University of Ferrara by the research group of Prof. Remo Guerrini. Concentrated solutions of ligands were made in distilled water (1 mM) and kept at −20 °C until use. Successive dilutions of ligands were made in Hank’s Balanced Salt Solution (HBSS) and HEPES buffer (containing 0.005% Bovine Serum Albumine (BSA)).

Cell Culture: Chinese Hamster Ovary (CHO) cells stably coexpressing the NOP opioid receptors and the G_αqi5_ protein were used. Cells were generated as described by Camarda and coworkers. They were cultured in Dulbecco’s Medium (DMEM) and Ham’s F-12 (1:1) supplemented with 10% fetal bovine serum (FBS), streptomycin (100 μg/mL), penicillin (100 IU/mL), l-glutamine (2 mmol/L), geneticin (G418; 200 μg/mL), fungizone (1 μg/mL), and hygromycin B (100 μg/mL). Cell cultures were kept at 37 °C in 5% CO_2_-humidified air. When confluence was reached, cells were detached by trypsinization, and 50,000 cells/well were seeded into 96-well black, clear-bottom plates 24 h before the test. The following day, after 24 h, the cells were used for testing.

Calcium Mobilization Assay: Calcium mobilization studies were performed using the automated fluorometer microplate reader FlexStation II (Molecular Devices, Sunnyvale, CA, USA), which was employed to detect changes in fluorescence intensity. At the assay time, cells were preincubated for 30 min at 37 °C protected from light with a loading solution consisting of HBSS supplemented with 2.5 mM probenecid, 3 μM Fluo-4 AM, and 0.01% pluronic acid. After that, the loading solution was discarded and 100 μL/well of Brillant Black solution (consisting of HBSS with 20 mM HEPES, 2.5 mM probenecid, and 500 μM Brilliant Black (Sigma-Aldrich, St. Louis, MO, USA)) was added and incubated for an additional 10 min into the fluorometer. After placing both plates (cell culture and compound plate) into the fluorometric imaging plate reader FlexStation II, fluorescence changes were measured. Serial dilutions of ligands were prepared in HBSS buffer with 20mM HEPES and 0.02% of BSA. Automated addition of peptides or vehicle solutions were automatically added in a volume of 50 μL/well. The present studies were performed at 37 °C to facilitate drug diffusion into the wells. Maximum change in fluorescence, expressed as percent over the baseline fluorescence, was used to determine agonist response. 

Data analysis and terminology: All data were analyzed using Graph Pad Prism 6.0 (La Jolla, CA, USA). Concentration–response curves were fitted using the four-parameter log–logistic equation. Data were expressed as mean ± sem of 5 experiments performed in duplicate. Agonist effects were expressed as the maximum change in percent over the baseline fluorescence. Baseline fluorescence was measured in wells treated with vehicles. Agonist potency was expressed as pEC_50_, which is the negative logarithm to base 10 of the agonist molar concentration that produces 50% of the maximal possible effect of that agonist.

### 3.3. Experimental Protocols

#### 3.3.1. Synthesis of methyl (S)-3-(4-hydroxy-3-nitrophenyl)-2-(picolinamido)propanoate (2)

To a solution of 4-Nitro-L-phenylalanine (1) (13.26 mmol, 3.00 g) in anhydrous MeOH (100 mL), SOCl_2_ (14.58 mmol, 1.06 mL) was added in a dropwise manner. The reaction mixture was heated at reflux overnight. The solvent was removed under vacuum to give the crude methyl-4-hydroxy-3-nitro phenylalanine hydrochloride, which was washed with 50 mL of saturated sodium bicarbonate aqueous solution (to pH~8) and extracted with DCM. The organic layers were combined and evaporated under vacuum to give the corresponding ester as yellowish solid, which was used directly for the next step. A mixture of the previous crude amino product, picolinic acid (15.39 mmol, 1.89 g), HATU (15.39 mmol, 5.85 g), and DIPEA (32.07 mmol, 5.59 mL) in 150 mL of DCM was stirred at room temperature overnight. Then, the reaction was quenched with a saturated NH_4_Cl aqueous solution and the two layers were separated. The aqueous layer was extracted with DCM (3 times), and the organic layers were combined, dried over Na_2_SO_4_, filtered, and concentrated in vacuum. The residue was purified by using flash chromatography (1:1 petroleum ether/AcOEt) to afford compound 2 (3.54 g) as an orange oil. Yield: 80% overall.

MS (ESI): *m/z* [M + H]^+^ calcd for C_16_H_15_N_3_O_6_ 346.31, found 346.31.

^1^H NMR (400 MHz, Chloroform-d) δ 10.47 (d, J = 0.4 Hz, 1H), 8.61–8.44 (m, 2H), 8.14 (dt, J = 7.8, 1.1 Hz, 1H), 7.91 (d, J = 2.2 Hz, 1H), 7.85 (td, J = 7.7, 1.7 Hz, 1H), 7.50–7.36 (m, 2H), 7.06 (d, J = 8.6 Hz, 1H), 5.06 (dt, J = 8.3, 6.0 Hz, 1H), 3.78 (s, 3H), 3.33–3.14 (m, 2H). ^13^C NMR (101 MHz, Chloroform-d) δ 171.39, 164.18, 154.31, 149.13, 148.52, 138.82, 137.53, 133.48, 128.74, 126.73, 125.42, 122.42, 120.34, 53.30, 52.77, 37.26.

#### 3.3.2. Synthesis of methyl (S)-3-(4-((tert-butyldimethylsilyl)oxy)-3-nitrophenyl)-2-(picolinamido)propanoate (3)

To a solution of 2 (10.25 mmol, 3.54g) in DCM (150 mL), imidazole (11.27 mmol, 0.76g) and TBSCl (11.27 mmol, 1.69 g) were added. The reaction mixture was stirred at room temperature overnight before the reaction was quenched by adding saturated NaHCO_3_. The organic layer was separated, and the aqueous layer was extracted (3 times) with DCM. The reunited organic layers were dried over Na_2_SO_4_, filtered, and concentrated under vacuum to give the TBS ether as a yellowish oil, and the crude was purified by using flash chromatography (3:2 petroleum ether/AcOEt) to afford compound 3 (3.72 g) as a yellowish oil. 

Yield: 79%

MS (ESI): *m/z* [M + H]^+^ calcd for C_22_H_29_N_3_O_6_Si 460.57, found 460.57. 

^1^H NMR (400 MHz, Chloroform-d) δ 8.56 (ddd, J = 4.8, 1.8, 1.0 Hz, 2H), 8.16 (dt, J = 7.8, 1.1 Hz, 1H), 7.86 (td, J = 7.7, 1.7 Hz, 1H), 7.62 (d, J = 2.3 Hz, 1H), 7.46 (ddd, J = 7.6, 4.8, 1.2 Hz, 1H), 7.31–7.23 (m, 1H), 6.88 (d, J = 8.5 Hz, 1H), 5.04 (dt, J = 8.3, 6.1 Hz, 1H), 3.76 (s, 3H), 3.31–3.14 (m, 2H), 0.98 (s, 9H), 0.22 (d, J = 1.3 Hz, 6H).

^13^C NMR (101 MHz, Chloroform-d) δ 171.42, 163.97, 149.01, 148.44, 148.28, 141.75, 137.76, 134.79, 129.46, 126.75, 126.27, 122.59, 122.40, 53.39, 52.71, 37.21, 25.64, 18.30, −4.26.

#### 3.3.3. Synthesis of methyl (S)-3-(3-amino-4-((tert-butyldimethylsilyl)oxy)phenyl)-2-(picolinamido)propanoate (4)

Compound 3 (8.09 mmol, 3.72 g) was dissolved in AcOEt (53.96 mL, 0.15 M) and set up in continuous-flow hydrogenator reactor H-Cube Pro Thales-Nano at 55 °C, 20 bar, in 0.5 mL/min flow, with Pd/C (10 mol%) as catalyst. When the reaction was completed, monitored via mass spectrometry, the solvent was concentrated in vacuum to obtain the crude product 4 as red-orange oil, which was purified by using flash chromatography (1:1 petroleum ether/AcOEt) to afford compound 4 (3.02 g) as an orange oil. 

Yield: 87%

MS (ESI): *m/z* [M + H]^+^ calcd for C_22_H_31_N_3_O_4_Si 430.59, found 430.59.

^1^H NMR (400 MHz, Chloroform-d) δ 8.54 (ddd, J = 4.8, 1.7, 0.9 Hz, 1H), 8.45 (d, J = 8.4 Hz, 1H), 8.15 (dt, J = 7.9, 1.1 Hz, 1H), 7.82 (td, J = 7.7, 1.7 Hz, 1H), 7.41 (ddd, J = 7.6, 4.8, 1.2 Hz, 1H), 6.64 (d, J = 8.0 Hz, 1H), 6.59 (d, J = 2.2 Hz, 1H), 6.45 (dd, J = 8.1, 2.2 Hz, 1H), 4.98 (dt, J = 8.3, 6.2 Hz, 1H), 3.71 (s, 3H), 3.32 (s, 2H), 3.08 (d, J = 6.3 Hz, 2H), 1.05–0.95 (m, 9H), 0.21 (s, 6H).

^13^C NMR (101 MHz, Chloroform-d) δ 172.15, 164.11, 149.51, 148.40, 142.26, 137.85, 137.35, 129.48, 126.42, 122.38, 119.54, 118.55, 116.74, 53.65, 52.39, 37.87, 25.95, 18.36, −4.13.

#### 3.3.4. Synthesis of methyl (S)-3-(4-((tert-butyldimethylsilyl)oxy)-3-(dibenzylamino)phenyl)-2-(picolinamido)propanoate (5)

To a solution of the aniline derivative 4 (7.03 mmol, 3.02 g) in CH_3_CN (100 mL), benzyl bromide (17.57 mmol, 2.09 mL) and potassium carbonate (14.06 mmol, 1.94 g) were added. The mixture was stirred at 80 °C overnight. The crude mixture was purified by using flash chromatography (1:1 AcOEt/petroleum ether) to afford 5 as an orange oil (3.04 g). 

Yield: 71%

MS (ESI): *m/z* [M + H]^+^ calcd for C_36_H_43_N_3_O_4_Si 610.84, found 610.84. 

^1^H NMR (400 MHz, Chloroform-d) δ 8.44 (d, J = 8.4 Hz, 1H), 8.38 (d, J = 4.8 Hz, 1H), 8.14 (dt, J = 7.8, 1.1 Hz, 1H), 7.80 (td, J = 7.7, 1.7 Hz, 1H), 7.34 (t, J = 6.3 Hz, 1H), 7.26–7.14 (m, 6H), 7.09 (d, J = 6.4 Hz, 4H), 6.78 (d, J = 8.0 Hz, 1H), 6.68 (d, J = 8.2 Hz, 1H), 6.56 (s, 1H), 4.93 (q, J = 6.5 Hz, 1H), 4.18 (d, J = 14.3 Hz, 2H), 4.08 (d, J = 14.3 Hz, 2H), 3.62 (s, 3H), 3.16–2.91 (m, 2H), 1.04 (s, 9H), 0.25 (s, 6H).

^13^C NMR (101 MHz, Chloroform-d) δ 171.90, 163.98, 149.53, 148.38, 147.88, 142.63, 138.24, 137.31, 129.21, 128.78, 128.17, 126.99, 126.38, 123.09, 122.34, 120.19, 54.19, 53.57, 52.31, 37.85, 26.11, 18.52, −3.89.

#### 3.3.5. Synthesis of methyl (S)-3-(4-((tert-butyldimethylsilyl)oxy)-5-(dibenzylamino)-2-methylphenyl)-2-(picolinamido)propanoate (6)

To a solution of compound 5 (4.98 mmol, 3.04 g) in toluene (90 mL) K_2_CO_3_ (14.95 mmol, 2.06 g), CH_3_I (24.92 mmol, 1.55 mL), and Pd(OAc)_2_ (0.49 mmol, 0.11 g) were added. The mixture was stirred at 120 °C overnight. After 24 h, the reaction was cooled to r.t. and filtered through celite pad and washed with AcOEt (50 mL). The filtrate was concentrated under vacuum to obtain the crude product. The crude product was purified by using flash chromatography (3:7 AcOEt/petroleum ether) to afford 6 (2.83 g) as a yellowish solid. 

Yield: 91%

MS (ESI): *m/z* [M + H]^+^ calcd for C_37_H_45_N_3_O_4_Si 624.87, found 624.87.

^1^H NMR (400 MHz, Chloroform-d) δ 8.43 (s, 1H), 8.39 (s, 1H), 8.12 (dt, J = 7.8, 1.1 Hz, 1H), 7.78 (td, J = 7.7, 1.7 Hz, 1H), 7.38–7.28 (m, 1H), 7.20 (ddd, J = 12.6, 7.7, 5.9 Hz, 6H), 7.13–7.04 (m, 4H), 6.65 (s, 1H), 6.51 (s, 1H), 4.87 (q, J = 7.2 Hz, 1H), 4.10 (d, J = 10.8 Hz, 4H), 3.59 (s, 3H), 3.09 (dd, J = 14.1, 6.4 Hz, 1H), 3.00 (dd, J = 14.1, 7.3 Hz, 1H), 2.25 (s, 3H), 1.03 (s, 9H), 0.25 (d, J = 2.3 Hz, 6H).

^13^C NMR (101 MHz, Chloroform-d) δ 172.36, 164.01, 149.41, 148.33, 147.66, 140.28, 138.40, 137.33, 130.56, 129.48, 129.20, 128.31, 128.10, 126.91, 126.82, 126.39, 123.93, 122.34, 54.36, 52.96, 52.34, 35.34, 26.11, 18.95, −3.88. MP 104–106 °C.

#### 3.3.6. Synthesis of methyl (S)-3-(5-amino-4-((tert-butyldimethylsilyl)oxy)-2-methylphenyl)-2-(picolinamido)propanoate (7)

Compound 6 (4.53 mmol, 2,83g) was dissolved in AcOEt (100 mL), and the reductive hydrogenation was performed by the addition of Pd/C (10 mol%) under H_2_ atmosphere. Once the reaction was completed, monitored via mass spectrometry, the solvent was concentrated in vacuum to afford the crude product, which was purified by using flash chromatography (1:1 AcOEt/petroleum ether) to obtain 7 (1.69 g) as a yellowish oil. 

Yield: 84%

MS (ESI): *m/z* [M + H]^+^ calcd for C_23_H_33_N_3_O_4_Si 444.62, found 444.62.

^1^H NMR (400 MHz, Chloroform-d) δ 8.54 (ddd, J = 4.8, 1.7, 0.9 Hz, 1H), 8.45 (d, J = 8.3 Hz, 1H), 8.12 (dt, J = 7.8, 1.1 Hz, 1H), 7.81 (td, J = 7.7, 1.7 Hz, 1H), 7.40 (ddd, J = 7.6, 4.7, 1.2 Hz, 1H), 6.57 (s, 1H), 6.56–6.48 (m, 1H), 4.93 (dt, J = 8.3, 7.0 Hz, 1H), 3.69 (s, 3H), 3.63 (d, J = 20.4 Hz, 2H), 3.10 (dd, J = 14.1, 6.9 Hz, 1H), 3.04 (dd, J = 14.0, 7.3 Hz, 1H), 2.21 (s, 3H), 0.99 (s, 9H), 0.21 (s, 6H).

^13^C NMR (101 MHz, Chloroform-d) δ 172.52, 164.10, 149.43, 148.34, 142.21, 137.34, 135.25, 127.49, 127.02, 126.41, 122.36, 120.72, 117.76, 53.04, 52.34, 35.47, 25.91, 18.84, 18.33, −4.13.

#### 3.3.7. Synthesis of (S)-2-((tert-butyldimethylsilyl)oxy)-5-(3-methoxy-3-oxo-2-(picolinamido)propyl)-4-methylbenzenediazonium (8)

To a solution of compound 7 (3.81 mmol, 1.69 g) dissolved in anhydrous THF (30 mL), cooled to -10 °C, iso-pentyl nitrite (7.61 mmol, 1.02 mL) and HBF_4_ (15.23 mmol, 0.95 mL) were added. The reaction was stirred for 4 h at -10 °C, and an orange precipitate was formed, which was directly used as the wet crude for the next step. A diazocopulation assay was performed on crude 8, with positive result. 

#### 3.3.8. Synthesis of methyl (S)-3-(4-((tert-butyldimethylsilyl)oxy)-2-methylphenyl)-2-(picolinamido)propanoate (9)

To a solution of Fe_2_SO_4_ (3.81 mmol, 0.57 g), 15 mL of DMF was added dropwise to the crude compound 8 (3.71 mmol, 1.73 g) solubilized in DMF (5.00 mL). The reaction was stirred overnight, at r.t. The solvent was removed under vacuum, and the residue was dissolved in DCM. The organic layer was washed with water, dried over Na_2_SO_4_, filtered, and concentrated in vacuum. The crude orange oil was purified by using flash chromatography (3:7 AcOEt/petroleum ether) obtaining compound 9 as a yellow solid (0.55 g).

Yield: 34%

MS (ESI): *m/z* [M + H]^+^ calcd for C_23_H_32_N_2_O_4_Si 429.60, found 429.60.

^1^H NMR (400 MHz, Chloroform-*d*) δ 8.56 (ddd, *J* = 4.8, 1.9, 0.9 Hz, 1H), 8.49 (d, *J* = 8.4 Hz, 1H), 8.14 (dt, *J* = 7.8, 1.2 Hz, 1H), 7.89–7.77 (m, 1H), 7.48–7.38 (m, 1H), 6.96 (d, *J* = 8.3 Hz, 1H), 6.65 (d, *J* = 2.6 Hz, 1H), 6.57 (dd, *J* = 8.2, 2.6 Hz, 1H), 4.97 (dt, *J* = 8.5, 7.2 Hz, 1H), 3.68 (s, 3H), 3.18 (dd, *J* = 14.1, 6.9 Hz, 1H), 3.12 (dd, *J* = 14.1, 7.3 Hz, 1H), 2.31 (s, 3H), 0.96 (s, 9H), 0.16 (s, 6H).

^13^C NMR (101 MHz, Chloroform-*d*) δ 172.28, 163.63, 154.50, 149.02, 147.87, 137.98, 137.64, 130.80, 127.17, 126.41, 122.52, 122.10, 117.45, 52.89, 52.21, 35.38, 25.66, 19.49, 18.17, −4.44. MP 106–108 °C.

#### 3.3.9. Synthesis of (S)-2-((tert-butoxycarbonyl)amino)-3-(4-hydroxy-2-methylphenyl)propanoic acid (10)

Once purified, compound 9 was dissolved in HCl 6N aqueous solution (21.81 mmol, 1.81 mL) and heated at 110 °C for 24 h. The obtained hydrolyzed crude product was concentrated under vacuum to reduce the volume, and the crude solution was used directly for the subsequent protection step.

The HCl salt just synthesized was directly used as crude and diluted in water/dioxane (1:2) solution (0.2 M). The mixture was basified with NaOH 2N aqueous solution until obtaining of pH value of 10/11 at 0 °C. Boc_2_O (1.53 mmol, 0.33 g) was added, and the reaction was left stirring at r.t. for 12 h. The completion of the reaction was monitored using ESI mass spectrometry and TLC. The dioxane was removed under vacuum, and HCl 1N aqueous solution was added at 0 °C to pH 1. The mixture was extracted with ethyl acetate (3 times), and the organic phases combined were dried over Na_2_SO_4_ and concentrated under vacuum. The crude was purified by using flash chromatography (7:3 AcOEt/petroleum ether/1% acetic acid) and crystallized with 2:1 Et_2_O/petroleum ether, obtaining a white solid as final product, 10. 

Yield: 45% (0.17 g white solid). Rt: 32.1 min. ee%: 100%.

HRMS *m/z*: [M − H]^−^ calcd for C_15_H_21_NO_5_ 294.1347, found 294.1348.

^1^H NMR (400 MHz, DMSO-d_6_) δ 12.47 (s, 1H), 9.06 (s, 1H), 7.06 (d, J = 8.4 Hz, 1H), 6.95 (d, J = 8.2 Hz, 1H), 6.54 (d, J = 2.6 Hz, 1H), 6.47 (dd, J = 8.2, 2.6 Hz, 1H), 3.99 (ddd, J = 10.0, 8.4, 4.6 Hz, 1H), 2.91 (dd, J = 14.2, 4.7 Hz, 1H), 2.67 (dd, J = 14.2, 10.0 Hz, 1H), 2.19 (s, 3H), 1.32 (s, 9H).

^13^C NMR (101 MHz, DMSO-d_6_) δ 174.37, 156.17, 155.93, 137.49, 131.17, 126.78, 117.22, 112.79, 78.42, 54.59, 33.78, 28.62, 19.47. MP 84–86 °C.

#### 3.3.10. Synthesis of (S)-2-((((9H-fluoren-9-yl)methoxy)carbonyl)amino)-3-(4-hydroxy-2-methylphenyl)propanoic acid (11)

Once purified, compound 9 was dissolved in HCl 6N aqueous solution (21.93 mmol, 1.82 mL) and heated at 110 °C for 24 h. The obtained hydrolyzed crude product was concentrated under vacuum to reduce the volume, and the crude solution was used directly for the subsequent protection step.

The HCl salt synthesized was used as the crude. It was diluted in water/dioxane (1:2) solution (0.2 M) and basified with Na_2_CO_3_ (3.88 mmol, 0.41 g) aqueous solution until pH 10/11 at 0 °C. FmocCl (1.16 mmol, 0.30 g) was added, and the reaction was left stirring at r.t. for 2 h. The reaction was monitored by using ESI mass spectrometry and TLC. The dioxane was then removed under vacuum, and HCl 1N aqueous solution was added at 0 °C to pH 1. The mixture was extracted with ethyl acetate (3 times), and the organic phases combined were dried over Na_2_SO_4_ and concentrated under vacuum. The crude was purified by using flash chromatography (1:1 AcOEt/petroleum ether/1% acetic acid) and crystallized with 2:1 Et_2_O/petroleum ether.

Yield: 62% (0.34 g white solid)

HRMS *m/z*: [M − H]^−^ calcd for C_25_H_23_NO_5_ 416.1503, found 416.1508.

^1^H NMR (400 MHz, DMSO-d_6_) δ 9.09 (s, 1H), 7.95–7.84 (m, 2H), 7.74–7.59 (m, 3H), 7.46–7.36 (m, 2H), 7.36–7.24 (m, 2H), 7.00 (d, J = 8.2 Hz, 1H), 6.55 (d, J = 2.6 Hz, 1H), 6.48 (dd, J = 8.2, 2.6 Hz, 1H), 4.18 (q, J = 4.7 Hz, 3H), 4.13–3.99 (m, 1H), 3.00 (dd, J = 14.2, 4.5 Hz, 1H), 2.73 (dd, J = 14.2, 10.2 Hz, 1H), 2.21 (s, 3H).

^13^C NMR (101 MHz, DMSO-d_6_) δ 173.68, 155.97, 155.77, 143.79, 140.67, 137.01, 130.79, 127.62, 127.29, 127.06, 125.26, 120.10, 116.84, 112.41, 65.62, 54.55, 46.58, 33.42, 19.05. MP 153–155 °C.

## 4. Conclusions

In conclusion, it was possible to obtain in good to moderate yield of (L)-2-methyl tyrosine, (L)-Mmt, through a novel pathway, different from the ones reported in the literature, removing possible byproducts from uncontrolled alkylation.

Moreover, the synthesis of this product turned out to be an affordable and low-cost synthesis, because it did not require complicated steps or expensive reagents. 

Even if the presence of a metasubstituted aromatic ring is widely known to represent a smart strategy to control the palladium-catalyzed alkylation leading to the only monosubstituted product, the choice of an easily and selectively removable bulky functional group, permissive for the Pd-catalyzed C(sp^2^)–H activation reaction, was not obvious.

It was possible to synthesize a few N/OFQ-related peptides, comparing the presence of Mmt and Dmt in place of the natural Phe^1^, which is present in N/OFQ and in the shorter fragment N/OFQ(1–13)-NH_2_. The four peptides were tested on a calcium mobilization assay in CHO_hNOP_ cells expressing the G_αqi5_ protein and showed good results in terms of potency; the occurring [Mmt^1^]N/OFQ(1-13)-NH_2_ was 10-fold more potent than the Dmt derivative and similar to N/OFQ(1-13)-NH_2_ and N/OFQ.

Finally, the present study describes the design, synthesis, and in vitro pharmacological evaluation of [Mmt^1^]N/OFQ(1–13)-NH_2_, which, thanks to our results, represents an intriguing possibility in terms of further exploration of highly potent synthetic peptides acting as NOP receptor agonists.

## Figures and Tables

**Figure 1 pharmaceuticals-16-01592-f001:**
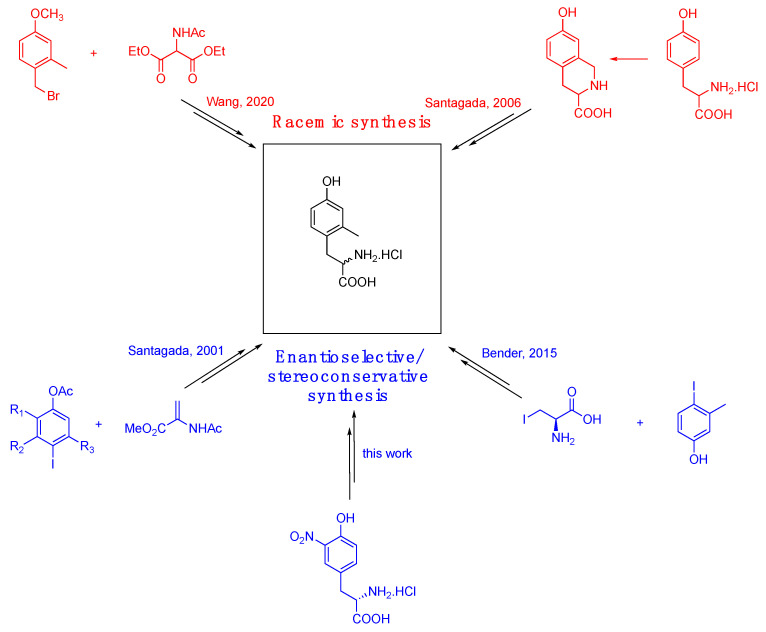
Brief comparison between previous Mmt syntheses [24,26,27,28] and this work.

**Figure 2 pharmaceuticals-16-01592-f002:**
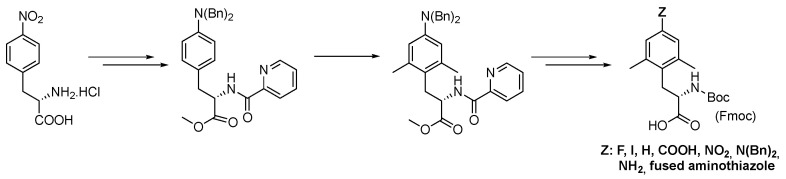
Dmt synthesis via Pd-catalyzed C(sp^2^)–H, Trapella, and Poli pathway.

**Figure 3 pharmaceuticals-16-01592-f003:**
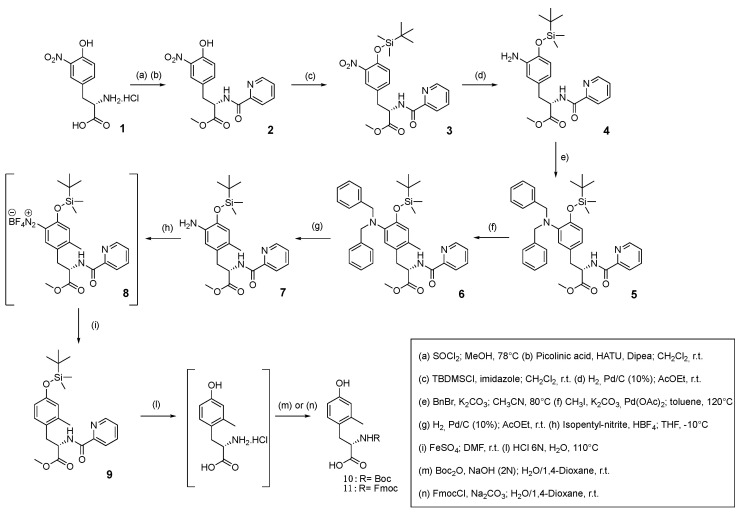
Traceless bulky regioselective synthesis of Mmt via Pd-catalyzed C(sp^2^)–H activation.

**Figure 4 pharmaceuticals-16-01592-f004:**
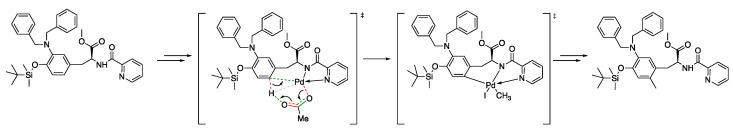
Proposed regioselective Pd-catalyzed C(sp^2^)–H sterically constrained monomethylation mechanism.

**Figure 5 pharmaceuticals-16-01592-f005:**
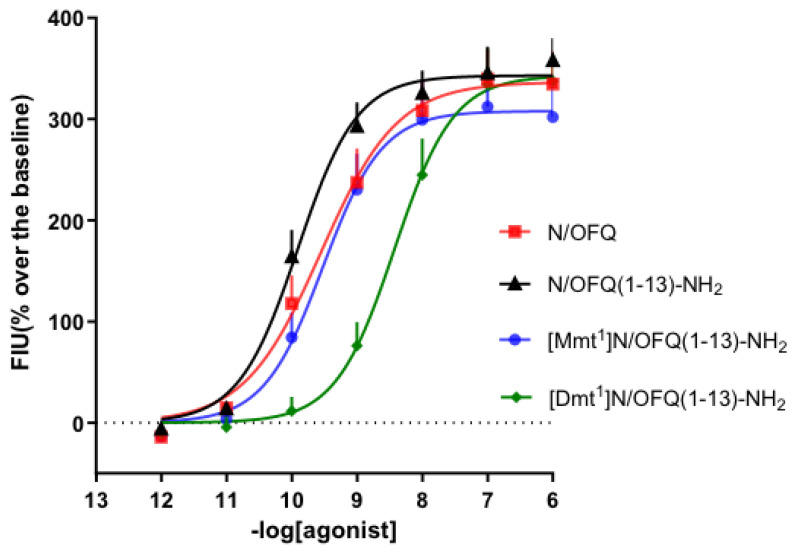
Calcium mobilization assay in CHO cells stably expressing the NOP receptor and the G_αqi5_ chimeric G protein. Concentration–response curves to N/OFQ, N/OFQ (1-13)-NH_2_, [Mmt^1^] N/OFQ (1-13)-NH_2_, and [Dmt^1^] N/OFQ (1-13)-NH_2_. Data are mean ± SEM of at least 6 separate experiments made in duplicate.

**Table 1 pharmaceuticals-16-01592-t001:** In vitro effects of the synthesized compounds in calcium mobilization studies performed on CHO cells coexpressing the NOP receptor and the Gα_qi5_ chimeric protein.

Calcium Mobilization in CHO_NOP + Gαqi5_ Cells		
	pEC_50_ (CL_95%_)	E_max_ + S.E.M.
N/OFQ	9.56 (9.02–10.09)	335 ± 33
N/OFQ(1-13)-NH_2_	9.82 (9.45–10.18)	358 ± 22
[Mmt^1^]N/OFQ(1-13)-NH_2_	9.47 (8.92–10.01)	302 ± 39
[Dmt^1^]N/OFQ(1-13)-NH_2_	8.35 (7.94–8.77)	337 ± 26

## Data Availability

Data are contained within the article and Appendix A.

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
