# Peer review of "(L)-Monomethyl Tyrosine (Mmt): New Synthetic Strategy via Bulky ‘Forced-Traceless’ Regioselective Pd-Catalyzed C(sp^2^)–H Activation"

_pharmaceuticals, 2023, doi:10.3390/ph16111592_

Round 1

Reviewer 1 Report

Comments and Suggestions for Authors

In the manuscript being reviewed, the authors detail the synthesis of (L)-Mono-methyl Tyrosine (Mmt) through a sterically forced regioselective Pd-catalyzed C(sp2)–H activation process. Notably, the use of dibenzyl amine as the bulky group stands out because of its easy removal or conversion capabilities, adding to the versatility of the method presented. While the synthesis pathway does involve multiple steps, it remains straightforward with simple reaction setup. New compounds are characterized by NMR and MS. The as-synthesized Mmt derivatives are ready to be inserted into peptide chains for bioactivity studies. Overall, I think that this is a good contribution to Pharmaceuticals, and it can be accepted after minor revision.

A few more comments:

1.        Page 4, Figure 4. The substituent on the phenyl ring should be OTBMS, not TBMS.

2.        Did the author check the ee value of the final products 10 and 11? Any racemization was observed as compared to the starting material 1?

3.        Besides the MS provided in the SI, the authors need to provide the HRMS of all the new compounds. Also, melting points are missing for solid compounds.

4.        In the SI, on page S10, please double-check the spectrum of “HPLC of compound 11 inserted in position 1 of N/OFQ(1-13)-NH2”.

5.        Additional recommended references, RSC Adv., 2020,10, 11013-11023; J. Med. Chem. 2006, 49, 6, 1882-1890; J.Peptide Sci. 2001, 7, 374-385.

Comments on the Quality of English Language

Minor editing needed.

Author Response

  1. The abbreviation has been modified as suggested by reviewer 1.
  2. We added the ee% value of (L)-Boc-Mmt. In the manuscript and the chromatograms in the SI. We didn’t insert this information at the beginning because we took inspiration from the work published by Zhang and Ma (Ref 28 of the manuscript), in which it was reported that the synthesis was stereoconservative, and from our point of view, the reaction conditions were very similar and so we thought we could assume this data was already noted. Anyway, we performed a chiral column on compound 10 to verify the absence of any racemization process, obtaining 100% of ee%.
  3. The HRMS of final compounds were added and MP of solid compounds.
  4. We checked the HPLC spectrum of compound 11, and in our word file is clear, the retention time is related with the peptide indicated. Maybe the resolution of manuscript is different once in the system of the Journal.
  5. We proceeded with the insertion of the recommended additional references, as suggested by reviewer 1.

Reviewer 2 Report

Comments and Suggestions for Authors

This manuscript describes the synthesis of 2-Me tyrosine through palladium catalyzed C-H activation/methylation of the C-2 hydrogen on the aryl ring using picolinamide directing group bound of the amino group of the amino acid.

Curiously the enantioselective synthesis of this material has been neglected despite its interest as surrogate of tyrosine and phenylalanine in various peptide. The present synthesis is quite lengthy with 11 steps but is selective and effective. The obtained amino acid has been transformed into its fmoc derivative and incorporated in a neuropeptide by the team of R. Guerrini. The N/OFQ neuropeptide incorporating the mono-Me Tyrosine instead of phenylalanine was found  as active as the native peptide and more active than the peptide having incorporated the more frequent di-Me tyrosine.

I think that synthesis of this material by Schöllkopf bis-lactim method or through enantioselective alkylation of glycine Schiff base might be a shorter synthetic route. Anyway the present synthesis works, the manuscript is clear and well written. I recommend its publication provided the authors address the following minor points.

1)      What about Pd catalyzed methylation of the nitro derivative 3 ? It may be expected that the electron withdrawing nitro group induces C-H abstraction at C-2 while the second reaction at C-6 will be much slower. This electronic instead of steric control could shorten the synthesis

2)      The integrity of the stereochemistry must be check through [a]D or chiral HPLC analysis, because many intermediate combining ester and the picolinamido group (3 to 9) may racemize during the process.

Author Response

1)As suggested by reviewer 2, we previously tried this synthetic pathway, but the nitro group on the aromatic ring seems to inhibit the Pd-catalyzed C(sp2)-H alkylation. As reported in our previous work, the presence of the nitro group in position 4 of the aromatic ring has been already observed to be completely inefficient toward this alkylation reaction, otherwise it would have been convenient for us to skip some reaction steps.

2)As reported to the answer of reviewer 1, we added the ee% value of (L)-Boc-Mmt in the manuscript and the chromatograms in the SI. We didn’t insert this information at the beginning because we took inspiration from the work published by Zhang and Ma (Ref 28 of the manuscript), in which it was reported that the synthesis was stereoconservative, and from our point of view, the reaction conditions were very similar and so we thought we could assume this data was already noted. Anyway, we performed a chiral column on compound 10 to verify the absence of any racemization process, obtaining 100% of ee%.

Reviewer 3 Report

Comments and Suggestions for Authors

Fantinati et al. have elegantly delineated a synthesis of (L)-mono-methyl tyrosine (Mmt) leveraging a Pd-catalyzed regioselective C(sp2)-H activation, which intriguingly is modulated by the steric hinderance of the NBn2 group. While this innovative approach to Mmt synthesis is commendable, its elongated 11-step process warrants exploration of more concise alternatives.

For a comprehensive comparison, it would be valuable if Figure 1 could incorporate the efficient 3-step method for Mmt synthesis, in which Negishi coupling is the key step, as documented in ACS Med. Chem. Lett. 2015, 6, 12, 1199–1203.

Author Response

We added in fig.1 (and in the main text) the brief synthetic pathway suggested by reviewer 3. The synthesis described is far shorter than the one presented in the present work, but the yield seems lower.

Reviewer 4 Report

Comments and Suggestions for Authors

Overall, the manuscript is well-prepared and presents valuable findings. However, there are a couple of issues that need to be addressed before it can be considered for publication:

1.   On Page 3, Lines 74 and 76, the abbreviation "MOP" is used without prior explanation. It is unclear to the reader what "MOP" stands for at this point in the manuscript. However, on Page 6, Line 163, the term "mu opioid receptor" is introduced, which seems to correspond to the abbreviation "MOP". For clarity, it would be advisable to introduce the abbreviation "MOP" alongside its full form "mu opioid receptor" upon its first mention in the manuscript, ensuring that readers are not left wondering about its meaning.

2.   Mass spectroscopy data format, m/z: [M + H]+ calcd for ~; found ~,

In the mass spectroscopy data section, that the decimal points are represented with commas, e.g., "346,31". The conventional representation in scientific literature is to use periods for decimal points. Thus, "346,31" should be corrected to "346.31". Please review the entire manuscript to ensure consistency in this notation.

3.   In the provided mass spectroscopy data, the format presented is "MS (ESI): m/z [M+H]+ 346,31". The standard representation in scientific literature should include both calculated and observed m/z values, and the use of periods for decimal points. Please correct the format to: "m/z: [M + H]+ calcd for ~; found 346.31". Additionally, ensure to provide the calculated value for a complete presentation of the data.

With these corrections, I believe the manuscript will be in excellent shape for publication. Great work, and I look forward to seeing the revised version.

Author Response

  1.  We proceeded by adding mu opioid receptor in brackets just after the first citation on MOP. See line 80 in the main text.
  2. All the value has been modified, as suggested by reviwer 3.

  3. All the value has been modified, as suggested by reviwer 3, adding found and calcd values.

Reviewer 5 Report

Comments and Suggestions for Authors

Authors:

I have reviewed your manuscript submitted to the journal Pharmaceuticals. I have found the manuscript fine. It contains all necessary parts. The synthesis is well done, and the compounds are characterized by the relevant analytical methods. The Introductions gives enough items of information on the state-of-the-art in the given topic. Results describe the synthetic procedure, as well as pharmacological investigation. Conclusions are consise and informative.

Formal point of view:

The stereochemistry descriptor "L" should be written in capital letter, not in italics. Please correct throughout the manuscrupt during the minor revision time.

Author Response

We changed “L” in italics to “L” in capital letter as suggested by Reviewer 5.